# Endoscopic Ultrasound-Guided Needle-Based Confocal Endomicroscopy as a Diagnostic Imaging Biomarker for Intraductal Papillary Mucinous Neoplasms

**DOI:** 10.3390/cancers16061238

**Published:** 2024-03-21

**Authors:** Shreyas Krishna, Ahmed Abdelbaki, Phil A. Hart, Jorge D. Machicado

**Affiliations:** 1Division of Gastroenterology, Hepatology and Nutrition, The Ohio State University Wexner Medical Center, Columbus, OH 43210, USA; krishna.111@osu.edu (S.K.); ahmed.abdelbaki@osumc.edu (A.A.);; 2Division of Gastroenterology and Hepatology, University of Michigan, Ann Arbor, MI 48109, USA

**Keywords:** endoscopic ultrasound, confocal endomicroscopy, IPMN, artificial intelligence, pancreatic cyst, pancreatic cancer

## Abstract

**Simple Summary:**

Pancreatic cysts represent a category of lesions that arise in the pancreas with the potential to transform into cancer. Intraductal papillary mucinous neoplasms (IPMNs) are the most common among these cysts. Current methods to detect early cancer in IPMNs are inaccurate. This review summarizes the latest advances in the diagnosis and risk assessment of IPMNs and mainly focuses on a novel imaging technique known as confocal laser endomicroscopy. Additionally, this review explores the integration of this imaging technique with artificial intelligence to optimize decision-making in managing these lesions.

**Abstract:**

Pancreatic cancer is on track to become the second leading cause of cancer-related deaths by 2030, yet there is a lack of accurate diagnostic tests for early detection. Intraductal papillary mucinous neoplasms (IPMNs) are precursors to pancreatic cancer and are increasingly being detected. Despite the development and refinement of multiple guidelines, diagnosing high-grade dysplasia or cancer in IPMNs using clinical, radiologic, endosonographic, and cyst fluid features still falls short in terms of accuracy, leading to both under- and overtreatment. EUS-guided needle-based confocal laser endomicroscopy (nCLE) is a novel technology that allows real-time optical biopsies of pancreatic cystic lesions. Emerging data has demonstrated that EUS-nCLE can diagnose and risk stratify IPMNs more accurately than conventional diagnostic tools. Implementing EUS-nCLE in clinical practice can potentially improve early diagnosis of pancreatic cancer, reduce unnecessary surgeries of IPMNs with low-grade dysplasia, and advance the field of digital pathomics. In this review, we summarize the current evidence that supports using EUS-nCLE as a diagnostic imaging biomarker for diagnosing IPMNs and for risk stratifying their degree of neoplasia. Moreover, we will present emerging data on the role of adding artificial intelligence (AI) algorithms to nCLE and integrating novel fluid biomarkers into nCLE.

## 1. Introduction

The increased resolution and utilization of cross-sectional imaging have led to a surge in the incidental detection of pancreatic cystic lesions. Approximately 10% of computed tomography (CT) scans and up to 45% of magnetic resonance imaging (MRI) studies in asymptomatic individuals detect pancreatic cystic lesions [1]. Among the different types of cysts, intraductal papillary mucinous neoplasms (IPMNs) are the most common and can potentially progress to malignancy. Although pancreatic intraepithelial neoplasia (PanIN) are the most frequent precursor lesions of pancreatic ductal adenocarcinoma (PDAC), comprising approximately 75–80% of all PDAC cases, these lesions are often not suitable for early radiologic detection. In contrast, up to 15% of all PDAC cases originate from IPMNs. Therefore, detection of IPMNs may offer the ability for early detection of high-grade dysplasia or adenocarcinoma in the pancreas [2].

IPMNs are broadly categorized into main duct and branch duct types based on their site of involvement. A mixed duct variety is identified when there is concurrent involvement of both the primary and branch ducts. Main duct IPMNs are associated with a higher risk of malignancy (~60% at ≥10 mm diameter), while branch-duct IPMNs have lower rates of prevalent cancer (~25% at ≥3 cm diameter) [3,4,5]. Surgical resection of branch-duct IPMNs is selectively indicated to treat lesions that are suspicious or have cytologic/histologic confirmation of high-grade dysplasia. However, the sensitivity for a cytologic or histologic diagnosis for high-grade dysplasia is very low in IPMNs, and the rate of false negative results can result in missed opportunities to treat early cancers. Therefore, we rely on different clinical biomarkers to stratify the risk of malignancy in IPMNs and decide between surgery or long-term surveillance [3,4,5]. Traditionally, this risk stratification has relied on cross-sectional imaging (CT, MRI), endoscopic ultrasound (EUS), and fine needle aspiration for fluid analysis of carcinoembryonic antigen (CEA), cytology, amylase, and glucose levels. However, these methods are suboptimal in diagnosing specific cyst types and stratifying the risk of malignancy in IPMNs [6,7]. Novel diagnostic modalities have emerged over the last decade and appear superior to traditional strategies for risk stratifying IPMNs. This narrative review will focus on the role of EUS-guided needle-based confocal laser endomicroscopy (EUS-nCLE) in diagnosing high-grade dysplasia and will also describe other emerging biomarkers.

## 2. Materials and Methods 

A comprehensive evaluation of the primary and secondary literature was conducted to explore the diagnosis and risk stratification of IPMNs using various methods. A thorough search was performed on PubMed, employing keywords such as “Confocal Laser Endomicroscopy”, “nCLE”, “endoscopic ultrasound”, “pancreas cyst”, “IPMN “, “IPMN risk stratification”, “Artificial intelligence”, and “Molecular Analysis”. Since EUS-nCLE for pancreatic cysts was not reported before 2009, the studies considered in the literature search spanned from January 2009 to December 2023. Original research as well as meta-analyses and reviews were included. 

The objective of this article was twofold: first, to dissect the diagnosis of IPMNs and other pancreatic cystic lesions employing a spectrum of modalities, with a particular focus on endoscopic ultrasound with confocal laser endomicroscopy. Second, to furnish a comprehensive overview of the efficacy of multiple modalities in discerning IPMNs with high-grade dysplasia or cancer from those with low-grade dysplasia. Through this detailed exploration, we aimed to illuminate the nuances of IPMN diagnosis and risk stratification, facilitating informed clinical decision-making in the management of these complex lesions.

## 3. Diagnosis of Mucinous Lesions

Accurate classification of pancreatic cystic lesions is crucial since mucinous lesions pose a higher risk of precancerous transformation than non-mucinous cysts. Intraductal papillary mucinous neoplasms (IPMNs) and mucinous cystic neoplasms (MCNs) fall under the mucinous category, whereas cystic neuroendocrine tumors, solid pseudopapillary neoplasms, and serous cystadenomas are non-mucinous lesions [8]. This review is focused on mucinous lesions, specifically branch-duct IPMNs. 

### Current Methods of IPMN Risk Stratification

The traditional diagnostic modalities used to evaluate pancreatic cysts, incorporated into the widely accepted consensus guidelines (Fukuoka Criteria and recent Kyoto Guidelines), diagnose the presence of high-grade dysplasia in IPMNs with an accuracy of 65–75% (low sensitivity but high specificity) [9,10]. As a result, approximately 60% of IPMNs undergoing a surgical resection for suspected high-grade dysplasia end up having only low-grade dysplasia, which translates into overtreatment and unnecessary surgeries (assuming there would not have been future malignant transformation) [11,12]. Notably, the rates of morbidity associated with pancreatectomies are high, including pancreatic duct leaks, bleeding, infection in the short term, as well as exocrine pancreatic insufficiency and diabetes mellitus in the long-term [13]. Conversely, traditional diagnostic approaches may miss or delay the diagnosis of invasive cancers during regular follow-ups for suspected branch-duct IPMNs. Therefore, there is an urgent need for highly accurate diagnostic tools that can effectively stratify the risk of branch-duct IPMNs. These tools should aim to prevent unnecessary surgical resections while also minimizing the risk of overlooking lesions with high-grade dysplasia or cancer [8,14,15].

## 4. Confocal Endomicroscopy and the Diagnosis of Mucinous Lesions

EUS-guided needle-based confocal endomicroscopy (nCLE) is an innovative technology that offers real-time microscopic imaging of the epithelium lining of pancreatic cystic lesions, enabling in vivo histopathology. This is performed by inserting an nCLE mini probe through a conventional 19-gauge fine needle aspiration needle, which is advanced under endosonographic guidance into the lesion of interest using a transgastric or transduodenal approach. Notably, specific types of cysts exhibit distinct endomicroscopy imaging patterns, allowing for differentiation and improved diagnostic accuracy compared to traditional techniques [16,17,18]. IPMNs show finger-like papillary projections with an inner vascular core. Mucinous cystic neoplasms are identified by horizon-type epithelial bands of variable thickness without papillary conformation. Cystic neuroendocrine tumors and solid pseudopapillary neoplasms display a trabecular pattern characterized by nests of cells separated by the stroma of the cyst [16,17,18]. Serous cystadenomas are distinguished by an intricate fern pattern of vascularity featuring a parallel or interconnected network of blood vessels. Finally, pseudocysts exhibit dark areas but lack a true epithelial lining and a vascular interstitium [16,17,18].

Recent studies have shown that both experienced operators in EUS-nCLE and novices recently trained in nCLE achieve high diagnostic accuracy and agreement on differentiating mucinous from non-mucinous pancreatic cysts and diagnosing specific cyst types [17,19,20]. In a recent study involving six endosonographers who were blinded to clinical data while reviewing expert-edited nCLE images from 29 patients (16 mucinous pancreatic cysts, 13 non-mucinous pancreatic cysts), the overall sensitivity, specificity, and accuracy for diagnosing mucinous pancreatic cysts were 95%, 94%, and 95%, respectively. Furthermore, the interobserver agreement and interobserver reliability were almost perfect [17]. Similar results have been observed in other studies summarized in a recent meta-analysis, in which EUS-nCLE had a sensitivity of 85%, a specificity of 99%, and a diagnostic accuracy of >95% to diagnose pancreatic cysts (mucinous vs. non-mucinous) [21]. Another larger study of 13 experts in EUS-nCLE who reviewed 76 edited nCLE videos reported similar results regarding the differentiation between mucinous pancreatic cysts and non-mucinous pancreatic cysts [19]. Moreover, this study evaluated the accuracy and reliability of EUS-nCLE for differentiating specific cyst types. The expert endosonographers diagnosed non-mucinous cysts with high accuracy (serous cystadenoma: 98%; cystic neuroendocrine tumor/solid pseudopapillary neoplasm: 96%; pseudocyst: 96%) and slightly less accuracy for mucinous lesions (IPMN: 86%; mucinous cystic neoplasm: 84%). The interobserver agreement was highest for serous cystadenoma (almost perfect; κ = 0.85), followed by IPMN (substantial, κ = 0.72), cystic neuroendocrine tumor/solid pseudopapillary neoplasm (significant, κ = 0.73), and at moderate levels for mucinous cystic neoplasm (κ = 0.47) and pseudocyst (κ = 0.57) [19]. A recent network meta-analysis suggested that EUS-nCLE achieved the highest accuracy in diagnosing pancreatic cysts compared to other advanced modalities, such as through needle biopsy and molecular analysis of cyst fluid [22]. 

Regarding training and performance of naïve nCLE imaging interpreters, a study compared didactic and self-directed teaching methods for learning nCLE patterns of 50 pancreatic cysts. Both teaching methods achieved high diagnostic accuracy (~95%) and substantial interobserver agreement in distinguishing mucinous from non-mucinous pancreatic cysts. These findings were consistent through varying levels of medical training (medical student to academic faculty) [20]. In a more recent study, which involved 21 early-career endosonographers naive to nCLE, a structured training program was given, followed by assessments using 80 expert-edited videos. The participants achieved competency in nCLE interpretation after a median of 38 subject video reviews with active feedback. Overall, the participants exhibited a remarkable 89% accuracy in distinguishing between mucinous and non-mucinous lesions, accompanied by substantial reliability. Notably, even eight weeks after nCLE training, the ability to independently differentiate mucinous from non-mucinous pancreatic cysts and accurately diagnose specific cyst types did not diminish, indicating the enduring nature of competency for nCLE interpretation [18].

## 5. Confocal Laser Endomicroscopy and Risk Stratification of IPMNs

Distinguishing branch-duct IPMNs with low-grade and high-grade dysplasia is crucial when deciding whether to proceed with surgical resection or remain on surveillance. Real-time in vivo histopathological assessment of the cyst epithelium with EUS-nCLE improves IPMN risk stratification compared to conventional methods, preventing oversight of IPMNS with high-grade dysplasia or cancer, while minimizing unnecessary resections of IPMNs with low-grade dysplasia.

Surgical histopathology findings indicate that low-grade dysplasia is characterized by a papillary configuration with a single layer of columnar cells exhibiting retained polarity. Conversely, high-grade dysplasia is characterized by a loss of nuclear polarity and an increased nuclear–cytoplasmic ratio [23]. Accordingly, EUS-nCLE imaging of IPMNs with low-grade dysplasia reveals a thin layer of epithelium, while those with high-grade dysplasia exhibit thicker and darker epithelium suggestive of cellular stratification (multiple layers) and nuclear abnormalities [24]. Figure 1 provides examples illustrating the differences between low-grade dysplasia and high-grade dysplasia. Additionally, nCLE imaging features such as papillary size and density, with larger dimensions and increased density, can further differentiate levels of dysplasia, with higher values indicating high-grade dysplasia. Furthermore, high-grade dysplasia in nCLE images may show distorted papillae, loss of finger-like architecture, absence of the lamina propria (vascular core), pronounced vascularity, and occasionally, large malignant cells (Figure 1). Continued research and validation are necessary to refine and establish the diagnostic utility of these nCLE imaging features in the clinical management of IPMNs [24].

## 6. Interobserver Studies for EUS-nCLE and Risk Stratification of IPMNs

In a study involving 26 IPMNs with 16 diagnosed as high-grade dysplasia based on surgical histopathology, multiple variables were assessed for the stratification of dysplasia, including papillary epithelial width, darkness, size, density, vascularity, and cellularity. Among the six observers with expertise in nCLE who qualitatively assessed the set of nCLE videos, the variables that showed the highest interobserver agreement and effectively identified high-grade dysplasia in the IPMNs were papillary epithelial width (sensitivity 90%, specificity 78%, accuracy 85%) and darkness (sensitivity 91%, specificity 73%, accuracy 84%). These two variables demonstrated moderate interobserver agreement (κ = 0.55) [24]. 

Using the same study population and set of videos, seven blinded nCLE-naïve observers underwent training and manually quantified the papillary width and darkness using dedicated software. Using this quantitative approach, the authors found that papillary width (cut-off ≥ 50 μm) and papillary darkness (cut-off ≤ 90 pixel intensity (lower levels being darker)) reached a maximum sensitivity and specificity of 88% and 100%, respectively [24]. A potential limitation of this approach is the partial reliance on the endosonographer to identify and precisely measure these structures, which is reflected by somewhat suboptimal interobserver agreement. Additionally, the risk stratification of branch-duct IPMNs relying on a quantitative image interpretation of papillary structures is laborious and time-consuming. Therefore, incorporating artificial intelligence and machine learning to identify and measure key imaging features would be beneficial for enhancing both the accuracy and reliability of branch-duct IPMN risk stratification with nCLE imaging [25].

## 7. Clinical Implications of EUS-nCLE

Recent meta-analyses have emphasized the value of incorporating nCLE into the decision-making processes for evaluating pancreatic cystic lesions [26,27]. This imaging biomarker has been demonstrated to be safe, with the major risk being post-procedure pancreatitis, which has been reported in 2% of patients [27]. In our experience, this risk can be mitigated by limiting the time of the procedure to no more than 10 min and administering rectal indomethacin. The use of nCLE has the potential to prevent unnecessary surgeries in patients with low-grade dysplasia who would have otherwise been taken for surgery due to radiologic changes of the cyst. A recent cost-benefit analysis found that EUS-nCLE prevents at least one unnecessary pancreatic surgery for every 10 subjects undergoing evaluation by current guidelines [28]. Not only can nCLE reduce unnecessary surgeries in patients with pancreatic cysts, but also can diagnose early malignancy (high-grade dyplasia or cancer), which allows for curative surgical resection and may potentially reduce the progression of untreated pancreatic cancer [29]. 

## 8. Limitations of EUS-nCLE

While confocal endomicroscopy holds promise, it also presents several limitations that must be addressed. Performing EUS-nCLE requires a special processor that is not universally available and is costly. The catheters required for nCLE can be re-used after special cleaning and processing; however, these probes have a finite lifespan. EUS-nCLE can only provide images from regions of the cyst wall to which the probe is in contact with during the exam. Therefore, given the rigid configuration of currently available EUS needles, nCLE is unable to produce images of the entirety of the cyst and some areas are left unexamined. Despite this, dysplastic transformation seems to develop in more than one area of the cyst and assessment of some representative areas may be sufficient [30]. Technological improvements of nCLE and EUS needles are needed to reduce some of these limitations. 

As nCLE is not widely available, EUS-nCLE is not routinely performed in many centers and is often not part of endoscopy training. Even when equipment is available, learning nCLE can be challenging as it requires mastering technical aspects and imaging interpretation skills. Achieving competency in diagnosing pancreatic cysts with nCLE requires at least 38 assessments [18]. However, this does not account on achieving technical dexterity, which requires a combination of workshops and hands-on procedures proctored by an expert in nCLE. Even among experts, there is still disagreement in image interpretation using the same set of images [19]. In hope of reducing these disagreements and further improving the diagnostic accuracy of this biomarker, ongoing research is trying to assess the role of applying artificial intelligence to nCLE or combining nCLE with other novel cyst fluid biomarkers. This will be discussed in the next two sections. Finally, there is a need for multicenter studies that can provide more robust data of applying this imaging biomarker in non-expert hands.

## 9. Applications of Artificial Intelligence for EUS and CLE

Artificial intelligence (AI) encompasses a spectrum of complexity, from machine learning to deep learning. In recent years, there has been a growing focus on applying deep learning techniques to enhance the risk stratification of pancreatic cystic lesions. Mainly, convolutional neural networks, specialized neural networks designed for image-based tasks, have emerged as a prominent deep learning algorithm in this field [25]. In recent years, advancements in AI have been employed in detecting, classifying, and diagnosing pancreatic cysts. AI empowers clinicians to delve into deeper layers of pancreatic cyst data that may not be discernible to humans. Integrating AI with novel diagnostic techniques like digital pathomics, radiomics, and genomics is paving the way for enhanced risk assessment of pancreatic cysts, offering promising possibilities for better understanding and stratification of IPMNs [25,31].

### 9.1. AI for Risk Stratification of IPMNs with EUS

AI has been applied in the risk stratification of IPMNs during EUS. In a study of 43 patients with IPMNs who underwent pancreatectomy, a model was trained using 3355 EUS images from these patients, resulting in the convolutional neural network classifying high-grade dysplasia with an accuracy of 99.6%. The AI model was superior in risk stratifying IPMNs than the guidelines, with accuracies ranging between 51.8% and 70.3% [32]. In a separate investigation involving 50 IPMNs and 3790 EUS images, a convolutional neural network model designed for IPMN risk assessment demonstrated impressive performance metrics. The deep learning program exhibited a sensitivity of 95.7%, a specificity of 96.2%, and an overall accuracy of 94.0% in predicting the malignant probability. In stark contrast, human interpretation yielded a diagnostic accuracy of only 56% [33]. 

### 9.2. AI for Risk Stratification of IPMNs with EUS-nCLE

In a recent study, two distinct convolutional neural network algorithms were utilized to classify branch-duct IPMNs as either low-grade dysplasia or high-grade dysplasia using EUS-nCLE [30]. The algorithms were trained and tested using expert-edited EUS-nCLE videos (15,027 video frames) from 35 consecutive patients with branch-duct IPMNs. One algorithm, called the segmentation-based model, was trained to detect and measure epithelial thickness and darkness in the images. The alternative algorithm, referred to as the agnostic holistic-based model, autonomously extracted nCLE features to facilitate risk stratification. When compared to the segmentation-based model and the holistic-based model, both of which were convolutional neural network algorithms, the performance of these models was evaluated in accordance with the guidelines laid out by the American Gastroenterological Association (AGA) and the Fukuoka International Consensus Guidelines. Both models demonstrated higher sensitivities for detecting high-grade dysplasia (holistic-based model: 83.3%; segmentation-based model: 83.3%; AGA: 55.6%; Fukuoka: 55.6%). They also exhibited increased overall accuracies (segmentation-based model: 82.9%; holistic-based model: 85.7%; AGA: 68.6%; Fukuoka: 74.3%) while maintaining comparable specificities (segmentation-based model: 82.4%; holistic-based model: 88.2%; AGA: 82.4%; Fukuoka: 94.1%). The study demonstrated that both the segmentation-based model and holistic-based model convolutional neural network algorithms achieved similar performance in risk stratification of IPMNs and improved accuracy compared to the AGA and Fukuoka guidelines. One of the critical limitations of this investigation involved using pre-edited nCLE videos to remove sequences deemed non-informative due to artifacts or redundancy. This potentially limits the accuracy of the convolutional neural network-AI algorithm and introduces a potential barrier to clinical implementation. Despite these limitations, this was the first study to highlight the potential of convolutional neural network-based approaches for enhancing the risk stratification of branch-duct IPMNs using nCLE imaging [30]. 

A scoping review identified 12 recent studies on AI models (encompassing radiomics and imaging), finding promising performances. However, a recurring theme was these studies’ relatively poor methodological quality [34]. The focus of future research should be on creating methodologically sound, generalizable, and clinically validated models. To provide genuine added value to clinical practice, these models should exhibit high sensitivity in excluding the presence of malignant potential and recommend surveillance for patients who would otherwise have been selected for resection. 

## 10. Novel Cyst Fluid Biomarker for Risk Stratification of IPMNs

### 10.1. Molecular Analysis

Molecular analysis of cyst fluid obtained via EUS-FNA has emerged as a viable option to determine the malignant potential of IPMNs [35]. Molecular analysis using next-generation sequencing is a powerful technique that involves collecting DNA from cyst fluid. The sequencing process consists in reading the DNA sequence at each fragment using various technologies. Identified genetic variations are then further analyzed to determine their clinical significance. This approach allows for the identification of various pancreatic cyst types, including serous cystadenomas, solid-pseudopapillary neoplasms, and cystic neuroendocrine tumors, which are characterized by specific mutations in the VHL, CTNNB1, and MEN1 genes, respectively. Mucinous cysts, on the other hand, frequently exhibit mutations in the mitogen-activated protein kinase (MAPK) gene, GNAS, and/or KRAS, diagnosing mucinous cysts with ~90% sensitivity and ~100% specificity [35,36,37].

Numerous studies have reported the high accuracy of next-generation sequencing in differentiating high-grade dysplasia and low-grade dysplasia among IPMNs [38,39,40]. In a recent prospective study of 1933 pancreatic cysts (251 with surgical histopathology), the combination of MAPK/GNAS and TP53/SMAD4/CTNNB1/mTOR had 88% sensitivity and 98% specificity in diagnosing advanced neoplasia [35]. Adding cytopathologic evaluation improved the sensitivity to 93%, while maintaining a high specificity of 95%. Fluid molecular analysis also holds promise in the prediction of future malignant progression in branch-duct IPMNs with low-grade dysplasia. Low-frequency mutations in the TP53, PIK3CA, and PTEN genes seem to precede histopathologic transformation into high-grade dysplasia [41]. A recent study revealed that KLF4 mutations may be a potential additional target for future risk stratification purposes [42]. 

Although molecular analysis of pancreatic cyst fluid has emerged as a valuable tool in combination with existing clinical management for enhancing the classification and risk stratification of IPMNs, it should be noted that molecular analysis alone has not yet been established as a reliable standalone diagnostic method for these lesions. The integration of EUS-nCLE, molecular analysis, and clinical data on an integrative AI algorithm can potentially increase the accuracy and reliability of risk stratification of IPMNs (Figure 2).

### 10.2. Telomere Fusions

The rapid cell division observed in neoplasia often leads to the shortening and potential fusion of telomeres, which are repetitive, non-coding DNA sequences. Telomere fusion-induced DNA damage can play a role in driving the progression of precancerous lesions. In a 2018 study involving 93 pancreatic cyst fluid samples, a telomere fusion assay demonstrated the absence of telomere fusions in branch-duct IPMNs with low-grad dysplasia. Moreover, the study reported a significant increase in the prevalence of telomere fusions in advanced versus low-grade lesions (*p* = 0.025) [43]. 

### 10.3. DNA Methylation

DNA methylation is a prevalent phenomenon in various types of human cancer, contributing to the aberrant silencing of genes. In a recent study involving 113 branch-duct IPMN cyst fluid specimens, the methylation patterns were analyzed for seven commonly hypermethylated genes (SOX17, PTCHD2, BNIP3, FOXE1, SLIT2, EYA4, and SFRP1). Among the various genetic markers examined, methylation of the SOX17 gene exhibited the most robust diagnostic potential in identifying advanced neoplasia in branch-duct IPMNs. It displayed a sensitivity of 83% and a specificity of 81.8%, making it a highly reliable indicator. Notably, the combination of hypermethylation in multiple genes demonstrated higher occurrence in branch-duct IPMNs with high-grade dysplasia compared to low-grade dysplasia lesions. A combination of DNA methylation targets achieved a diagnostic accuracy of 88%, surpassing the results obtained through individual marker analysis. This approach not only enhanced sensitivity and specificity but also improved the overall effectiveness of the diagnostic process [44]. In another study, two methylated genes (TBX15 and BMP3) were identified as valuable indicators for distinguishing between high-grade and low-grade IPMNs. When analyzed together, these genes showed enhanced discriminatory power, resulting in a sensitivity of 90% and specificity of 92%. This finding highlights the potential of utilizing multiple biomarkers to improve the accuracy of IPMN classification [45].

### 10.4. miRNAs and lncRNAs

MicroRNAs and long non-coding RNAs have also been identified as key markers in the risk stratification of branch-duct IPMNs [46]. MicroRNAs are short, non-coding RNA segments involved in post-transcriptional gene expression regulation, while lncRNAs play a significant role in gene expression regulation. In a study involving 12 IPMNs, elevated levels of six microRNAs (miR-711, miR-3679-5p, miR-6126, miR-6780b-5p, miR-6798-5p, and miR-6879-5p) were observed in the pancreatic cyst fluid of branch-duct IPMNs with high-grade dysplasia [47]. In another study focusing on long non-coding RNAs, lower levels of the RNA TUSC-7 were found to be significantly associated with pancreatic cystic lesions that progress to cancer [48]. 

### 10.5. Mass Spectrometry

Another tool helpful in the risk stratification of IPMNs is mass spectrometry, a technique that identifies molecules through the parameters of molecular weight, structure, and chemical formula. It does this through measuring the mass-to-charge ratio of ions. It assists in the analysis of protein profiles of cysts that could demonstrate high-grade dysplasia. In a diagnostic study conducted in 2018, researchers utilized targeted mass spectrometry analysis of pancreatic cyst fluid and focused on protein biomarkers derived from mucin-5AC and prostate stem-cell antigen. The study demonstrated an impressive accuracy of 96% in identifying advanced neoplasia [49]. 

## 11. Conclusions

EUS-nCLE demonstrates superior accuracy compared to conventional methods for both diagnosing IPMNs and stratifying their degree of dysplasia. However, this imaging biomarker poses some challenges in terms of interobserver agreement, platform design, clinical implementation and training. Adding machine learning and other AI models to EUS-nCLE can overcome some of the challenges with nCLE interpretation. Integrating novel diagnostic modalities such as cyst fluid next-generation sequencing and other emerging molecular markers to EUS-nCLE holds promise in more accurately risk stratifying IPMNs. Further validation, expansive patient data, and long-term follow-up are required to establish the efficacy and reliability of EUS-nCLE, either alone or combined with other methods, on diagnosing and risk stratifying IPMNS.

## Figures and Tables

**Figure 1 cancers-16-01238-f001:**
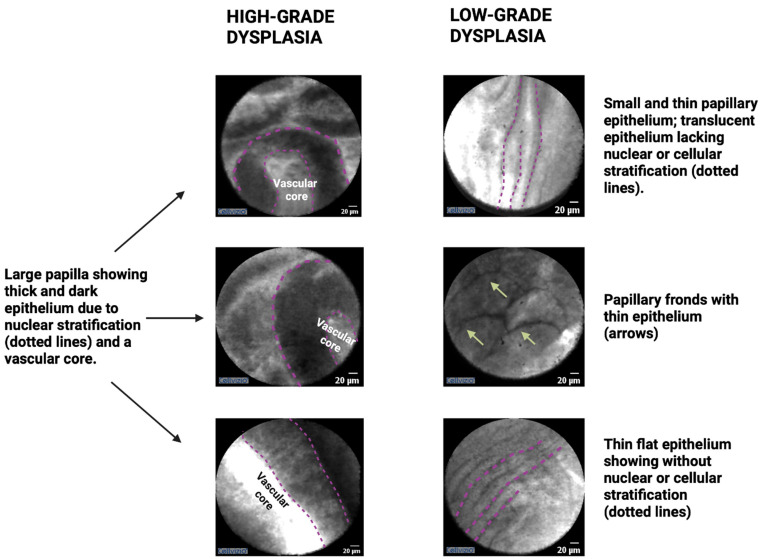
EUS-guided confocal endomicroscopy patterns of IPMNs with high-grade dysplasia or cancer (**left panel**) and low-grade dysplasia (**right panel**). Note: the figure is courtesy of the Division of Gastroenterology, Hepatology and Nutrition, The Ohio State University Wexner Medical Center.

**Figure 2 cancers-16-01238-f002:**
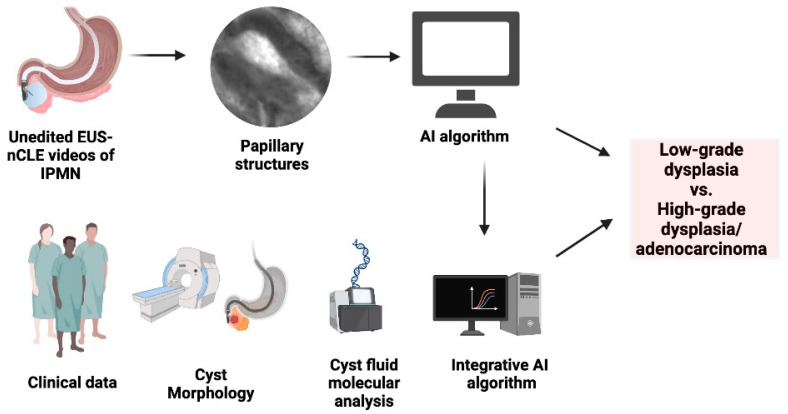
Integrated AI-driven risk stratification flowchart for predicting advanced neoplasia in IPMNs. Note: the figure is courtesy of the Division of Gastroenterology, Hepatology and Nutrition, The Ohio State University Wexner Medical Center.

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
