# Peer review of "Endoscopic Ultrasound-Guided Needle-Based Confocal Endomicroscopy as a Diagnostic Imaging Biomarker for Intraductal Papillary Mucinous Neoplasms"

_cancers, 2024, doi:10.3390/cancers16061238_

Round 1
Reviewer 1 Report
Comments and Suggestions for Authors
To improve the scientific quality of the manuscript, the authors should consider the following points:
1. Clarity and structure:
The manuscript lacks a clear and concise structure, making it difficult for readers to follow the flow and the huge amount of information. It would be beneficial to reorganize the content into distinct sections under clear headings. Describe the flow of all issues considered in the "Materials and Methods" section by stating the objectives of this narrative review, the method used for conducting their search, and the main search analysis. In this way, readers are provided with a rational and logical explanation of the issues considered, and the engine and MeSH terms used in this review .
2. Overreliance on acronyms:
The manuscript contains a disturbing overuse of acronyms, which could impede understanding if the reader is unfamiliar with the terms. The authors should consider limiting this overuse.
3. Incomplete citations:
Some statements lack proper citations to support claims. Authors should ensure that all assertions are backed by relevant references to strengthen their scientific reliability.
4. Inadequate discussion of limitations:
The manuscript fails to thoroughly discuss the limitations of the proposed methods and findings. Authors should provide a comprehensive analysis of potential biases, confounding factors, and technical limitations associated with the techniques discussed.
5. Insufficient Discussion of Clinical Implications:
While the manuscript touches upon the potential clinical implications of the proposed diagnostic methods, it lacks depth in discussing how these advancements may impact patient care, treatment strategies, and outcomes. Authors should provide a more robust discussion of the practical implications of their research in clinical settings.
6. Need for further validation:
Many of the novel techniques and biomarkers discussed are still in the early stages of development and require further validation through larger-scale studies and clinical trials. Authors should clearly acknowledge the preliminary nature of the evidence presented and emphasize the need for additional validation before widespread clinical implementation.
7. Improvement of figures:
Figures could be improved and comprehension could be facilitated, if possible.
8. Grammar and language:
Grammar errors should be checked by authors before submitting their manuscripts. English style can be improved.
9. Conclusion:
The conclusion could be strengthened by summarizing the key findings and implications more succinctly. Authors should also highlight future directions for research in the field and articulate clear recommendations for clinical practice based on current evidence.
Comments on the Quality of English Language8. Grammar and language:
Grammar errors should be checked by authors before submitting their manuscripts. English style can be improved.
Author Response
Greetings,
we addressed the first reviewers comments and answered the questions down below.
- Clarity and structure:
The manuscript lacks a clear and concise structure, making it difficult for readers to follow the flow and the huge amount of information. It would be beneficial to reorganize the content into distinct sections under clear headings. Describe the flow of all issues considered in the "Materials and Methods" section by stating the objectives of this narrative review, the method used for conducting their search, and the main search analysis. In this way, readers are provided with a rational and logical explanation of the issues considered, and the engine and MeSH terms used in this review.
Answer: Using this comment, we have made a few improvements. Firstly, we added a material and methods section right under the introduction. In that section, we included the methods used to search for literature, and also a brief overview of the flow and objectives of the narrative review. Additionally, we separated a few larger sections into different headings in order to conform to the flow previously established in the materials and methods.
- Overreliance on acronyms:
The manuscript contains a disturbing overuse of acronyms, which could impede understanding if the reader is unfamiliar with the terms. The authors should consider limiting this overuse.
Answer: We have eliminated most acronyms. PCL, HGD-Ca, LGD, SPN, MCN, DL, and ML were all expanded out using a “command find search and replace function”. We kept the abbreviation of IPMN though,
- Incomplete citations:
Some statements lack proper citations to support claims. Authors should ensure that all assertions are backed by relevant references to strengthen their scientific reliability.
Answer: We fixed this by adding citations to unbacked claims. These were usually facts implied by a previously (and continuously) cited paper, so we just repeated citations in most areas. In other areas we attempted to add them.
- Inadequate discussion of limitations:
The manuscript fails to thoroughly discuss the limitations of the proposed methods and findings. Authors should provide a comprehensive analysis of potential biases, confounding factors, and technical limitations associated with the techniques discussed.
Answer: We added a limitations section to delve deeper into the discussion of limitations. We mention all of the things that reviewer 1 points out as being helpful for the limitation section.
- Insufficient Discussion of Clinical Implications:
While the manuscript touches upon the potential clinical implications of the proposed diagnostic methods, it lacks depth in discussing how these advancements may impact patient care, treatment strategies, and outcomes. Authors should provide a more robust discussion of the practical implications of their research in clinical settings.
Answer: We added another section dedicated to clinical implications and included patient care, treatment strategies and outcomes. This covers the need for a more robust discussion on these issues.
- Need for further validation:
Many of the novel techniques and biomarkers discussed are still in the early stages of development and require further validation through larger-scale studies and clinical trials. Authors should clearly acknowledge the preliminary nature of the evidence presented and emphasize the need for additional validation before widespread clinical implementation.
Answer: We added a couple of lines to discuss the need for additional validation in the conclusions section.
- Improvement of figures:
Figures could be improved, and comprehension could be facilitated, if possible.
Answer: Figures were modified and improved. Specifically, we have included helpful visuals on the images of cysts themselves to make it easier to perceive and comprehend.
- Grammar and language:
Grammar errors should be checked by authors before submitting their manuscripts. English style can be improved.
Answer: Grammar errors were fully checked and parts of the paper that weren’t very concise were fixed to be more readable.
- Conclusion:
The conclusion could be strengthened by summarizing the key findings and implications more succinctly. Authors should also highlight future directions for research in the field and articulate clear recommendations for clinical practice based on current evidence.
Answer: We added a part in the conclusion to articulate recommendations for clinical practice. We discuss implications and the need for further validation as well.
Reviewer 2 Report
Comments and Suggestions for Authors
This is a review article regarding endoscopic ultrasound-guided needle based confocal endomicroscopy as a diagnostic imaging biomarker for intraductal papillary mucinous neoplasms.
The paper is interesting and clinically relevant. Manuscript is well written and organized. References are adequate and recent. It contains good quality figures. In figure 1, findings / images of EUS-guided confocal endomicroscopy are presented. Therefore, the figure legend should contain a source of images. It has to be added.
Minor editing of English language is required.
Author Response
This is a review article regarding endoscopic ultrasound-guided needle based confocal endomicroscopy as a diagnostic imaging biomarker for intraductal papillary mucinous neoplasms.
The paper is interesting and clinically relevant. Manuscript is well written and organized. References are adequate and recent. It contains good quality figures. In figure 1, findings / images of EUS-guided confocal endomicroscopy are presented. Therefore, the figure legend should contain a source of images. It has to be added.
Answer: A source of images in the figure was added. All of them are from the Ohio State University.
Reviewer 3 Report
Comments and Suggestions for Authors
1) General comments
Dr. Krishna and Dr. Machicado, et al. reviewed “Endoscopic Ultrasound-Guided Needle Based Confocal Endomicroscopy as a Diagnostic Imaging Biomarker for Intraductal Papillary Mucinous Neoplasms”. This article is well presented. The reviewer has some comments.
1. In this review article, the authors describe EUS-nCLE is useful for the diagnosis of IPMN and will be further developed with the help of AI. Please discuss the current limitations of EUS-nCLE, as well as complications and side effects of the procedure.
Author Response
1) General comments
Dr. Krishna and Dr. Machicado, et al. reviewed “Endoscopic Ultrasound-Guided Needle Based Confocal Endomicroscopy as a Diagnostic Imaging Biomarker for Intraductal Papillary Mucinous Neoplasms”. This article is well presented. The reviewer has some comments.
1. In this review article, the authors describe EUS-nCLE is useful for the diagnosis of IPMN and will be further developed with the help of AI. Please discuss the current limitations of EUS-nCLE, as well as complications and side effects of the procedure.
Answer: We discuss both of these aspects (limitations and complications) in the newly added “Limitations” section and the “Implications” section. These have new cited studies to back up their credibility as sections too.
Round 2
Reviewer 1 Report
Comments and Suggestions for Authors
The Authors satisfactorily addressed all reviewer's questions and criticisms.
Author Response
Thanks
Reviewer 3 Report
Comments and Suggestions for Authors
1) General comments
Dr. Krishna and Dr. Machicado, et al. revised “Endoscopic Ultrasound-Guided Needle Based Confocal Endomicroscopy as a Diagnostic Imaging Biomarker for Intraductal Papillary Mucinous Neoplasms”. This article is well presented.
Thank you for your reply. I read your responses for my questions. Your answers are precisely good, and I understood your points of view in your study.
Author Response
Thanks